# Analysis of the Properties of 44 ABC Transporter Genes from Biocontrol Agent *Trichoderma asperellum* ACCC30536 and Their Responses to Pathogenic *Alternaria alternata* Toxin Stress

**Hua-Ying Du** [1,†], **Yu-Zhou Zhang** [2,†,‡], **Kuo Liu** [1], **Pei-Wen Gu** [1], **Shuang Cao** [1], **Xiang Gao** [1], **Zhi-Ying Wang** [2], **Zhi-Hua Liu** [2,3] **and Ze-Yang Yu** [1,2,*]

1 School of Agriculture, Ningxia University, 489 Helan Mountain West Road, Yinchuan 750021, China
2 School of Forestry, Northeast Forestry University, 26 Hexing Road, Harbin 150040, China
3 College of Forestry, Shenyang Agricultural University, 120 Dongling Road, Shenyang 110866, China
* Correspondence: yzy@nxu.edu.cn; Tel.: +86-951-5015825; Fax: +86-951-5032599
† These authors contributed to the work equally and should be regarded as co-first authors.
‡ Current Address: Ningxia Forest Pest Control and Quarantine Station, 60 Nanxun West Road, Yinchuan 750021, China.

**Abstract:** ATP-binding cassette (ABC) transporters are involved in transporting multiple substrates, such as toxins, and may be important for the survival of *Trichoderma* when encountering biotic toxins. In this study, genome searching revealed that there are 44 ABC transporters encoded in the genome of *Trichoderma asperellum*. These ABC transporters were divided into six types based on three-dimensional (3D) structure prediction, of which four, represented by 39 ABCs, are involved in transport and the remaining two, represented by 5 ABCs, are involved in regulating translation. The characteristics of nucleotide-binding domain (NBD) are important in the identification of ABC proteins. Even though the 3D structures of the 79 NBDs in the 44 ABCs are similar, multiple sequence alignment showed they can be divided into three classes. In total, 794 motifs were found in the promoter regions of the 44 *ABC* genes, of which 541 were cis-regulators related to stress responses. To characterize how their ABCs respond when *T. asperellum* interact with fungi or plants, *T. asperellum* was cultivated in either minimal media (MM) control, C-hungry, N-hungry, or poplar medium (PdPap) to simulate normal conditions, competition with pathogens, interaction with pathogens, and interaction with plants, respectively. The results show that 17 of 39 transport ABCs are highly expressed in at least one condition, whereas four of the five translation-regulating ABCs are highly expressed in at least one condition. Of these 21 highly expressed *ABCs*, 6 were chosen for RT-qPCR expression under the toxin stress of phytopathogen *Alternaria alternata*, and the results show *ABC01*, *ABC04*, *ABC05*, and *ABC31* were highly expressed and may be involved in pathogen interaction and detoxifying toxins from *A. alternata*.

**Keywords:** biological control; ABC protein; detoxification

## 1. Introduction

Many *Trichoderma* species, such as *T. harzianum*, *T. asperellum*, and *T. virens*, are biocontrol agents which are widely distributed in nature and are used to protect crop plants in the form of commercial fungicides [1]. The resistance mechanisms of *Trichoderma* against phytopathogens include mycoparasitism, induction of resistance genes, niche exclusion competition, and plant growth promotion [2,3]. *Trichoderma* spp. can also protect stored fruits and flowers against pathogens [4]; hence, they are considered promising and environmentally friendly agents for controlling plant diseases affecting cereals as well as vegetable crops [5]. However, to perform their biocontrol role, *Trichoderma* spp. must first survive in hostile environments in which they are exposed to pesticides, pathogen toxins, and phytoalexin. ATP-binding cassette (ABC) proteins play an important role in this process.

ABCs were first discovered in bacteria but are now known to be ubiquitous, occurring in all organisms. The hallmark of ABC transporters is their ability to transport molecules such as toxins, ions, and proteins through membranes and into and out of cells in an energy-consuming process involving ATP hydrolysis via an ATP-binding cassette [6]. In most uptake systems, ABCs include a solute-binding protein, known as an extracytoplasmic receptor, which is lacking in export systems. Additionally, some ABCs are involved in regulating RNA translation or DNA repair rather than transport processes [7]. Full-size ABC transporters form a four-core domain structure formed from a single polypeptide chain [8] comprising two ATP-binding cassette nucleotide-binding domains (NBDs) and two hydrophobic transmembrane domains (TMDs). NBDs are characterized by highly conserved motifs found in most ABC families, including the Walker A motif (G-X(4)-G-K-[ST]), the Walker B motif ([hydrophobic](4)-[D]), and the ATP-binding motif ([LIVMFY]-S-[SG]-G-X(3)-[RKA]-[LIVMYA]-X-[LIV-MF]-[AG]) [9]. NBDs provide energy for the transport process. TMDs usually include six transmembrane-spanning ÿ $\alpha$-helices but can include 10, 17, or 19 ÿ $\alpha$-helices, and they determine transport function and substrate recognition. Half-sized ABCs form functional proteins by combining with other half-sized ABCs via homo- or heterodimerization [10].

ABC transporters are involved in exporting drugs or toxins, excreting endogenous metabolites, detoxification, and regulating growth [11], and they play an important role in pathogenic microorganisms in agriculture. In the bacterial phytopathogen *Agrobacterium tumefaciens* Smith & Townsend, ABCs import sugars, and sugar binding to the periplasmic binding protein ChvE triggers a signaling response resulting in the expression of virulence genes [12]. Furthermore, in the phytopathogenic fungus *Aspergillus nidulans* G. Winter, and AtrBp can prevent intracellular accumulation of toxicants and increase energy-dependent efflux [13]. In the phytopathogenic fungus *Magnaporthe grisea* M.E. Barr, expression of ABCs is induced by antifungal compounds, and the disruption of expression decreases viability and pathogenicity [14]. In another phytopathogenic fungus, *Botrytis cinerea* C. H. Persoon, disruption of the ABC transporter gene *BcatrB* increases sensitivity to a grapevine phytoalexin [15]. Thus, it is clear that ABC transporters are crucial for pathogenic microorganism survival and virulence. Upon investigating the interaction between phytopathogens and biocontrol fungi, researchers showed that *Trichoderma* species can outcompete phytopathogens and might possess the ability to detoxify toxins from plants or phytopathogens. Indeed, work on *Trichoderma atroviride* P. Karsten showed that the ABC transporter *hex1* in *T. atroviride* can improve tolerance to dichlorvos [16]. Another ABC transporter, taabc2, affects mycelial growth, spore formation, and pigment biosynthesis in *T. atroviride* [17], and taabc2 knockout reduces the antagonistic activity against the pathogen *Fusarium oxysporum* Schlecht. emend. Snyder & Hansen [18]. However, there are insufficient studies on ABC transporters in other *Trichoderma* species, and further knowledge on their potential contribution to biocontrol or detoxification is needed.

In this study, in order to determine the properties of ABC proteins in *T. asperellum* and further identify potential robust ABC proteins that can help *T. asperellum* survive under biotic stress, 44 ABC transporter genes were identified in the *T. asperellum* genome and characterized based on sequence analysis, structure analysis, functional prediction, and promoter analysis. Further, transcriptome- and transcription-level analysis of six genes under toxin conditions revealed which genes potentially respond to biotic stress. The findings expand our understanding of *Trichoderma* tolerance to pathogenic toxins or phytotoxins, and the characteristics of these potential genes can serve as a reference for identifying ABC genes or finding robust ABC genes in other *Trichoderma* species or organisms, which could ultimately assist with the development of improved environmentally friendly biocontrol agents.

## 2. Materials and methods

### 2.1. Analysis of 44 ABC Transporter Genes in the Trichoderma asperellum CBS433.97 Genome

All ABC transporter gene sequences from the genome of *T. asperellum* CBS433.97 were obtained via keyword searching (http://genome.jgi.doe.gov/pages/tree-of-life.jsf, accessed on 3 March 2021). Protein molecular weight was predicted using the Sequence Manipulation Suite (http://www.bioinformatics.org/sms/, accessed on 9 March 2021), and proteins were ranked according to size from large to small. Conserved domains were predicted using the BLAST program and analyzed using Pfam (http://pfam.sanger.ac.uk/, accessed on 9 March 2021). All non-ABC transporter genes were omitted. The isoelectric point (pI) was predicted using the ProtParam tool (http://web.expasy.org/protparam/, accessed on 10 March 2021).

### 2.2. Multiple Sequence Alignment of 79 NBDs from the 44 ABCs of T. asperellum CBS433.97 and Phylogenetic Analysis of 44 ABCs

Highly conserved motifs were observed in most NBDs of ABCs and are important characteristics of NBDs, so a total of 79 NBDs were identified among the 44 ABC transporter genes and analyzed using Clustal Omega (http://www.ebi.ac.uk/services/proteins, accessed on 10 March 2021). Sequence logos were drawn using WebLogo (http://weblogo.berkeley.edu/logo.cgi, accessed on 10 March 2021).

To investigate the 44 ABCs, a phylogenetic tree was constructed for the amino acid sequences of 44 ABCs using the maximum-likelihood method (model: LG + G + I + F) with 1000 bootstrap replicates in the MEGA 7.0 program.

### 2.3. Prediction of 3D Structure and Promoter of 44 ABCs in T. asperellum CBS433.97

The 3D structures of the 44 ABC transporters in *T. asperellum* were predicted using the automodel method in Swiss-Model (https://www.swissmodel.expasy.org/, accessed on 13 March 2021). Motifs within promoter regions were predicted using the SCPD promoter analysis website (http://rulai.cshl.edu/SCPD/, accessed on 17 March 2021).

### 2.4. The Expression Analysis of ABC Transporter Genes under Different Media Based on RNA-Seq

*Trichoderma asperellum* ACCC30536 (obtained from the Agricultural Cultural Collection of China, inoculum amount: $1 \times 10^7$ spores) was cultured with shaking (180 r/min) at 25 °C for 2 d in potato dextrose (PD) broth medium, and mycelia were harvest using a sterile gauze and washed with sterile purified water under aseptic conditions, and the mycelia were then subjected to four different culture media: mineral medium (MM) as used in a previous study [19], C-hungry medium (MM with 0.5% *w/v* ammonium sulfate), N-hungry medium (MM with 0.5% *w/v* glucose), and PdPap (*Populus davidiana* × *P. alba* var. *pyramidalis*) medium (MM containing 1% *w/v* root powder, 1% *w/v* stem powder, or 1% *w/v* leaf powder from PdPap). After culturing with shaking (180 r/min) at 25 °C for 72 h, the mycelia were washed with sterile purified water and transferred into centrifuge tubes, which were stored at −70 °C until they were processed for total RNA extraction using TRIzol reagent (Invitrogen, Carlsbad, CA, USA, Kit) [20]. The total amount of 3 μg mRNA

was obtained and fragmented into short fragments, followed by cDNA synthesis. The resulting library was sequenced using Illumina HiSeqTM 2000. The raw data were filtered, with the remaining reads referred to as "clean reads", which were aligned to reference genes and a genome file with SOAPaligner/SOAP2. The alignment data were utilized to calculate the distribution of reads on reference genes and perform coverage analysis. Quality control of alignment was then required to determine whether the samples needed resequencing. Qualified data were then subjected to several analyses, including gene expression, Gene Ontology (GO) and Kyoto Encyclopedia of Genes and Genomes (KEGG) annotation, alternative splicing, novel transcript prediction and annotation, and SNP analysis. Finally, four transcriptome libraries were constructed [21]. The gene expression level was calculated using the reads per kilobase of transcript per million mapped reads (RPKM) method. The dist method was used to determine the Euclidean distance, and the complete method includes the method for hierarchical clustering of ABC genes according to their expression.

### 2.5. Transcription Analysis of Six ABCs in Response to Toxins of Alternaria alternata CFCC82114

The poplar leaf blight strain *A. alternata* CFCC82114 (obtained from the China Forestry Culture Collection Center) was cultured with shaking (180 r/min) at 25 °C for 12 d in PD broth medium, the fermentation was filtered using a 2.5 μm filter, and the filtrate was stored at −80 °C. *Trichoderma asperellum* ACCC30536 ($1 \times 10^7$ spores/mL) was cultured with shaking (180 r/min) at 25 °C for 2 d in PD broth medium, the mycelia were washed and transferred to MM and toxin-induced medium (MM with 10% fermentation filtrate) under aseptic conditions with harvesting at 0, 6, 24, and 48 h for RNA extraction. The filamentous total RNA was extracted using the CTAB method, digested with DNaseI (Promega, Madison, WI, USA), and reverse-transcribed into cDNA using the PrimeScript RT Kit (Takara, Japan) according to the manufacturer's instructions. The transcription level was determined using RT-qPCR and was calculated according to the $2^{-\Delta\Delta Ct}$ method, using cDNA as template, and *actin*, *α-tubulin*, and *β-tubulin* as reference genes [20]. Three RT-qPCR replicates were analyzed per cDNA sample, and the primers were designed using Primer premier 6.0 software (Table 1). The experiment was set up with three replicates. The statistical significance of differences was determined using an independent-samples *t*-test ($p < 0.05$).

**Table 1.** Primers for RT-qPCR analysis.

| Gene Name | Primer Name | Sequence (5′–3′) | T$_m$ (°C) | Size of Product (bp) |
|---|---|---|---|---|
| *ABC01* | ABC1-L | TGCTGAAGAAGCCAACCGTCAT | 58.8 | 250 |
| | ABC1-R | CTTGTCTCGCACATGCTCCTGA | 59.0 | |
| *ABC02* | ABC2-L | GCTGATCCGAGGCGATCTGAAG | 59.2 | 248 |
| | ABC2-R | TGCGTGTGGAGAAGGAACTGAG | 54.5 | |
| *ABC04* | ABC4-L | CTGCTTCAACCGCCTCTCCAA | 58.8 | 231 |
| | ABC4-R | GCGTGCTGTCAGTGAACTCCT | 58.8 | |
| *ABC05* | ABC5-L | GACACTGGCTCTGTTCCGACTC | 59.0 | 246 |
| | ABC5-R | CCACCATCCATGCTGGCTACAT | 58.7 | |
| *ABC30* | ABC30-L | CCACCAACCTCGCTCCAACAT | 58.8 | 221 |
| | ABC30-R | CGTCGGCATCAATTCGTCCATC | 58.5 | |
| *ABC31* | ABC31-L | AGCGTCGTGTCATTGGCATCA | 58.9 | 258 |
| | ABC31-R | CGCAGTGGCTCTCAATCTCCTT | 58.7 | |
| *α-tubulin* | α-tu-L | CACATGGTTGACTGGTGCCCTA | 58.6 | 240 |
| | α-tu-R | CTCGCCCTCTTCCATACCCTCT | 59.0 | |
| *β-tubulin* | β-tu-L | TTCTTCCACCTTTGTCGGCAACTCCT | 59.1 | 227 |
| | β-tu-R | CCTCGTACTCCTCACCATCA | 59.1 | |
| *actin* | Act-L | GGCTCAGTCTAAGCGTGGTATCC | 59.0 | 251 |
| | Act-R | ACAGAACGGCCTGGATGGAGA | 59.0 | |

## 3. Results

### 3.1. Analysis of the 44 ABC Transporters in T. asperellum CBS433.97

The properties of the 44 ABC transporter genes identified in the *T. asperellum* CBS433.97 genome were analyzed (Table S1). ABC transporter genes were distributed on 17 different scaffolds, with the number of genes on each scaffold ranked from 1 to 7, and scaffold six includes the largest number (7). These genes contain different numbers of introns, ranging from 0 to 15; 17 genes (39%) have more than 4 introns, while 4 (9%), 13 (30%), 5 (11%), and 5 (11%) genes have 3, 2, 1, and 0 introns, respectively. The isoelectric point (pI) ranges from 5.48 to 9.87; 17 ABCs are basic (pI > 7) and 27 are acidic (pI < 7). Their molecular weight ranges from 55.33 to 187.56 kDa. Based on NCBI multidomain prediction, the 44 ABC transporter genes were classified according to nine different multidomains: ATM1 (4), CFTR (6), MdlB (5), MDR (8), MRP (9), Protein White (4), Rim protein (2), Uup (5), and 3a0123 (1). The 44 ABCs were divided into 11 different types based on the order of conserved domains. Furthermore, while most ABCs included both NBDs and TMDs, two proteins contain only NBDs, indicating that they are involved not in transport, but rather in translational regulation.

### 3.2. Multiple Sequence Alignment of the 79 NBDs Present in the 44 ABCs in T. asperellum CBS433.97

A total of 79 NBDs are found in the 44 ABC transporter proteins in *T. asperellum*. The Walker A, Walker B, and ATP-binding consensus motifs of the NBDs were analyzed. Walker A and Walker B motifs form the catalytic core domain and participate in nucleotide binding and phosphate/$Mg^{2+}$ coordination, respectively. The ATP-binding consensus motif is unique to ABC proteins and also interacts with ATP. The multiple sequence alignment and the Walker A motif (G-XX-[GA]-X-G-K-[ST]), ATP-binding consensus motif ([LIVMFYC]-S-[SGQDEVAHL]-G-X(3)-[RKALV]-[LIVMYATC]-X-[LIVMF]-[AGCT]), and Walker B motif ([ΦHY]-[ΦCY]-[ΦC]-[ΦCT]-[D]-[DEI]) are shown in Figure 1.

The multiple sequence alignment shows the conserved amino acids (marked by shade) were different; they were mainly concentrated in the eighth and ninth amino acids in "Walker A" and the sixth, eighth, twelfth, and thirteenth amino acids in the "ATP-binding motif". The sequences were divided into three main classes according to these differences. Based on the functional prediction based on 3D structure and the annotation of the conserved domain, sequences in the Class 1 (top) include those classified with "Protein White" and "Rim protein" multidomain. Sequences in Class 2 (middle) are mostly aligned with the "Uup" multidomain classification, the proteins of which are reported to be involved in translational regulation. Meanwhile, most of the sequences in Class 3 (bottom) are proteins predicted to be involved in efflux (MRP, CFTR, MDR MdlB, and ATM1 multidomain). Class 3 can be further divided into three subclasses based on the different characteristics of conserved residues (marked by shadow). Furthermore, we found that the NBDs were clustered at the same position: the first NBD clustered together and the second NBD were clustered together, which is marked by shadowing of the name.

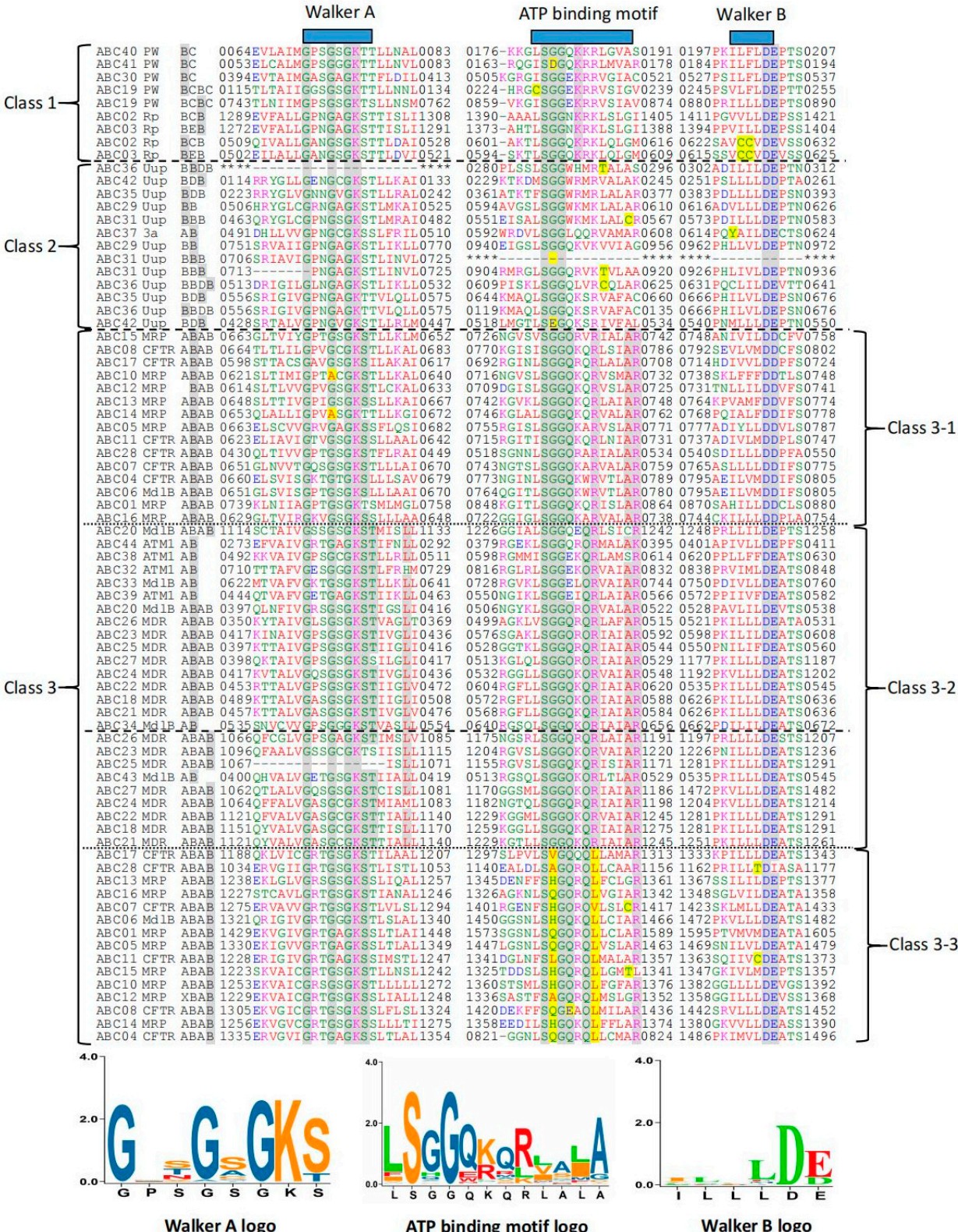

**Figure 1.** Alignment of the 79 NBDs within the 44 ABCs. NBDs were predicted using NCBI BlastP, and amino acid sequences that differ from previous studies are highlighted in yellow. Numbers in names indicate the ABC protein number. The B with the shadow in the names represent the NBD and the position of the peptide.

### 3.3. Phylogenetic Analysis of the 44 ABC Proteins

The phylogenetic relationships of 44 ABCs were investigated in further detail (Figure 2). Their distribution in the phylogenetic tree is closely correlated with the multidomain. The MRPs and CFTRs clustered in Clade I, the ATM1s and MDRs clustered in Clade II, and the Uups clustered in Clade VI. Except for ABC06, which was distributed in Clade I, other MdlBs were distributed in Clade II. However, the ABCs in "Protein White" do not have close and stable phylogenetic relationships with each other, and they were distributed in different clades. ABC30, ABC40, and ABC41 have closer evolutionary relationships with the proteins in Clades I and II, while ABC19 has a closer evolutionary relation with Uups (Clade VI). The ABC30 was the only protein in Clade III, and it does not have close phylogenetic relationships with the proteins in Clades I, II, and IV. In Clade IV, we found that two "Rim protein" and two "Protein White" members had close evolutionary relationships.

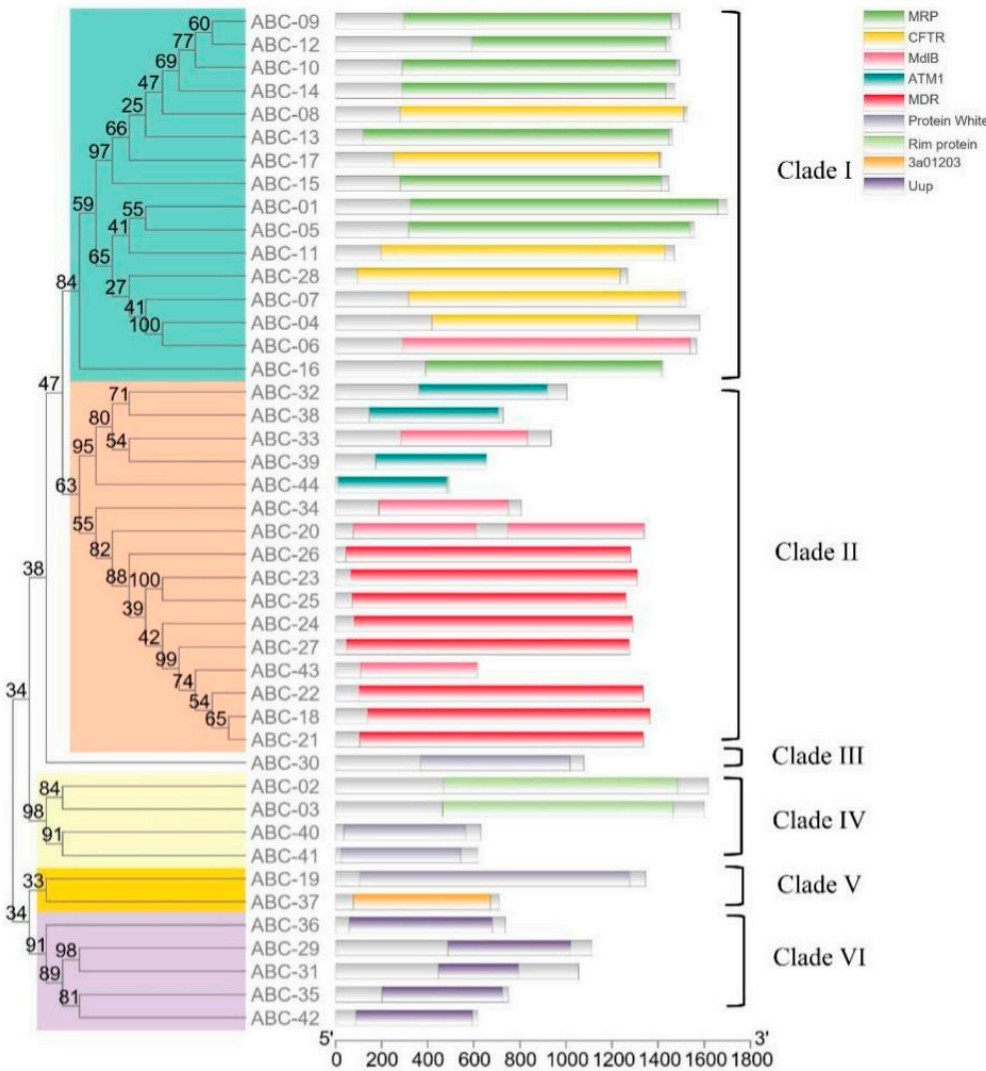

**Figure 2.** Phylogenetic analysis of the 44 ABC proteins. The phylogenetic tree was constructed using the maximum-likelihood method (model: LG + G + I + F) with 1000 bootstrap replicates in the MEGA 7.0 program. The positions of multidomains were drawn in the TBtools program.

### 3.4. Prediction of the 3D Structures of the 44 ABC Transporters in T. asperellum CBS433.97

The 44 ABC transporters in *T. asperellum* were divided into six types based on the predicted 3D structures, shown in different colors (Figure 3). The first four types (ABC-3D-1 to ABC-3D-4) are involved in transmembrane transport, and the other two (ABC-3D-5 and ABC-3D-6) are involved in translational regulation.

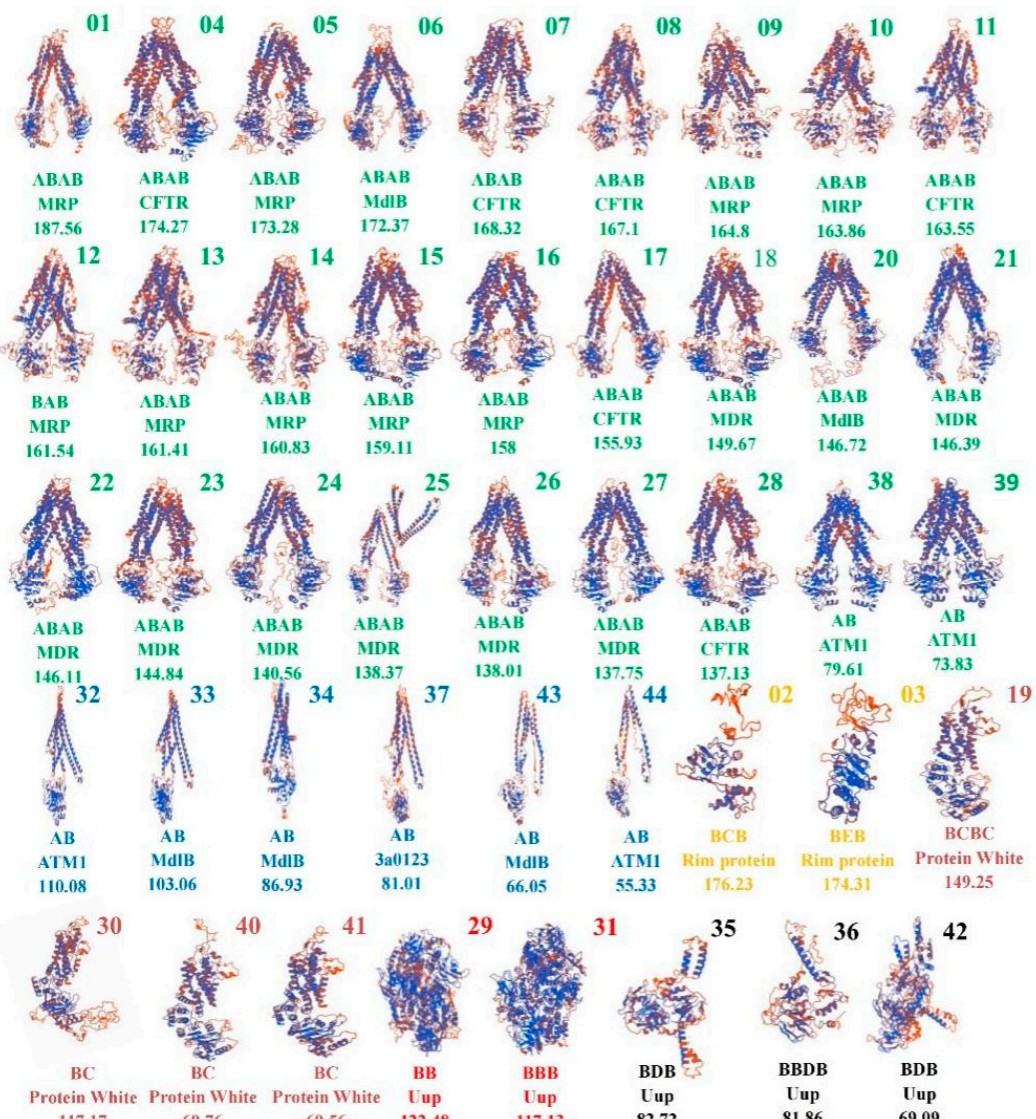

**Figure 3.** The predicted three-dimensional structures of the 44 ABC transporters in *T. asperellum*. The prediction was conducted using Clustal Omega (http://www.ebi.ac.uk/services/proteins, accessed on 13 March 2021). The number at the top right corner shows the protein number. The first line below the structure shows the order of conserved domains (A–E represents domains belonging to the ABC-membrane, ABC-ATPase, ABC2-membrane, ABC-tran-2, and acyl-transfer-3 superfamilies, respectively). The second line below the structure shows the multidomain type. The third line below the structure shows the protein molecular weight (kDa).

Based on Table 2 and Figure 3, we found that the ABC-3D-1 transport protein class (green in Figure 3) includes 27 of the 44 ABCs, all of which share a similar structure based on two pseudo-symmetric halves formed from a single polypeptide, with each half formed from several transmembrane helices (TMD), the NBD, and the various interconnecting loops and short helices. Proteins in the ABC-3D-2 class (blue in Figure 3) correspond with the half structures in the ABC-3D-1 class and must dimerize with another protein to form a functional transporter. The two proteins in the ABC-3D-3 class (yellow) belong to the multidomain category "Rim protein". The structure of ABC-3D-3 proteins consists of two NBD regulatory domains (RDs) and one TMD. The four proteins in the ABC-3D-4 class (pink in Figure 3) belong to the multidomain "Protein White", including both TMDs and NBDs, and the TMDs all belong to the ABC2-membrane superfamily. The ABC-3D-1 and ABC-3D-2 members account for 75% of all ABC transporters, indicating their importance in *T. asperellum.* Members of ABC-3D-5 (red in Figure 3) and ABC-3D-6 (dark in Figure 3) belong to the multidomain "Uup", the proteins in ABC-3D-5 only have NBDs while the proteins in ABC-3D-6 have TMDs, and the TMDs in ABC-3D-6 all belong to "ABC-tran-2" superfamily.

**Table 2.** Relation among structure, multidomain, and conserved domain arrangement order.

| Three-Dimensional Structure | Multidomain | Conserved Domain Arrangement Order |
|---|---|---|
| Structural type 1 | MRP, CFTR, MDR, MdlB | A-B-A-B |
| | ATM1 | A-B |
| Structural type 2 | ATM1, MdlB, 3a0123 | A-B |
| Structural type 3 | Rim protein | B-C-B, B-E-B |
| Structural type 4 | Protein White | B-C-B-C, B-C |
| Structural type 5 | Uup | B-B, B-B-B |
| Structural type 6 | Uup | B-D-B, B-B-D-B |

Note: In conserved domain arrangement order, A represents ABC-membrane superfamily, B represents ABC-ATPase superfamily, C represents ABC2-membrane superfamily, D represents ABC-tran-2 superfamily, and E represents acyl-transfer-3 superfamily.

### 3.5. Analysis of the Promoters of the 44 ABC Transporter Genes in T. asperellum CBS433.97

The 1000 bp upstream sequence of each of the 44 *ABC* transporter genes was cloned. The motifs in the promoter regions were predicted and located using the SCPD program (Figure 4). A total of 794 motifs of 39 different types were identified across the 44 promoters. In each promoter, motif numbers ranged from 9 to 32 and the type of motif ranged from 5 to 14. Of these, 541 motifs, including GCR1, GCN4, ADR1, STRE, HSTF, PHO4, GC/FAR, and ABF1, are closely related to stress responses. The three most abundant motifs are GCN4, GCR1, and ADR1, which are involved in metabolism and cell defenses, and 165, 93, and 67 motifs, respectively, were found for these. Furthermore, "ATM1" ABCs have an average of 21 motifs, which is the most, and "Rim protein" ABCs have an average of 13 motifs, which is the fewest. The motifs in "Uup" ABCs are the most diverse, with each "Uup" ABC having an average of 11 kind of motifs, while the "Rim protein" ABCs only have an average of 7 kinds of motifs (the fewest). Genes with the greatest number of motifs belonged to the "MRP" protein category, and the gene with the most motif types belonged to that of "Uup" protein (Figure 4).

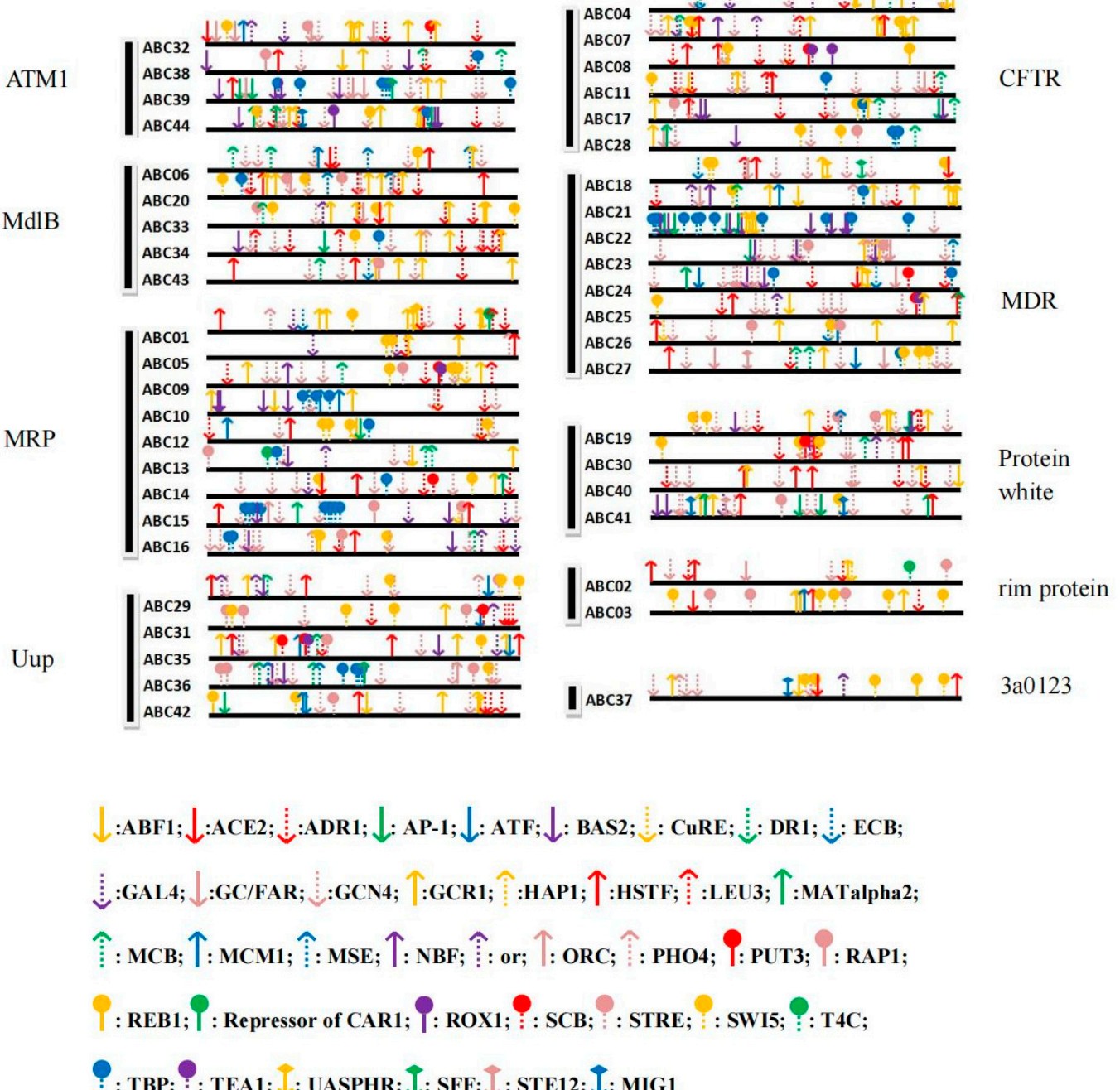

**Figure 4.** Regulatory element-binding motifs in the promoters of the 44 *ABC* transporter genes in *Trichoderma asperellum*.

*3.6. The Expression of 44 ABC Transporter Genes in T. asperellum ACCC30536 under Four Different Conditions*

The four transcriptomes were constructed, and the data were deposited in the SRA database (https://www.ncbi.nlm.nih.gov/sra, accessed on 26 August 2021) under the accession number PRJNA758014. Next, we examined the expression of ABC transporter genes in these four conditions: MM, C-hungry, N-hungry, and PdPap media (Figure 5). We found that *ABC31* and *ABC42* were the most highly expressed in all four conditions. *ABC31* expression was highest (59214 RPKM) in PdPap and lowest (8537 RPKM) in MM, whereas *ABC42* expression was highest (11790 RPKM) in N-hungry and lowest (3957 RPKM) in MM.

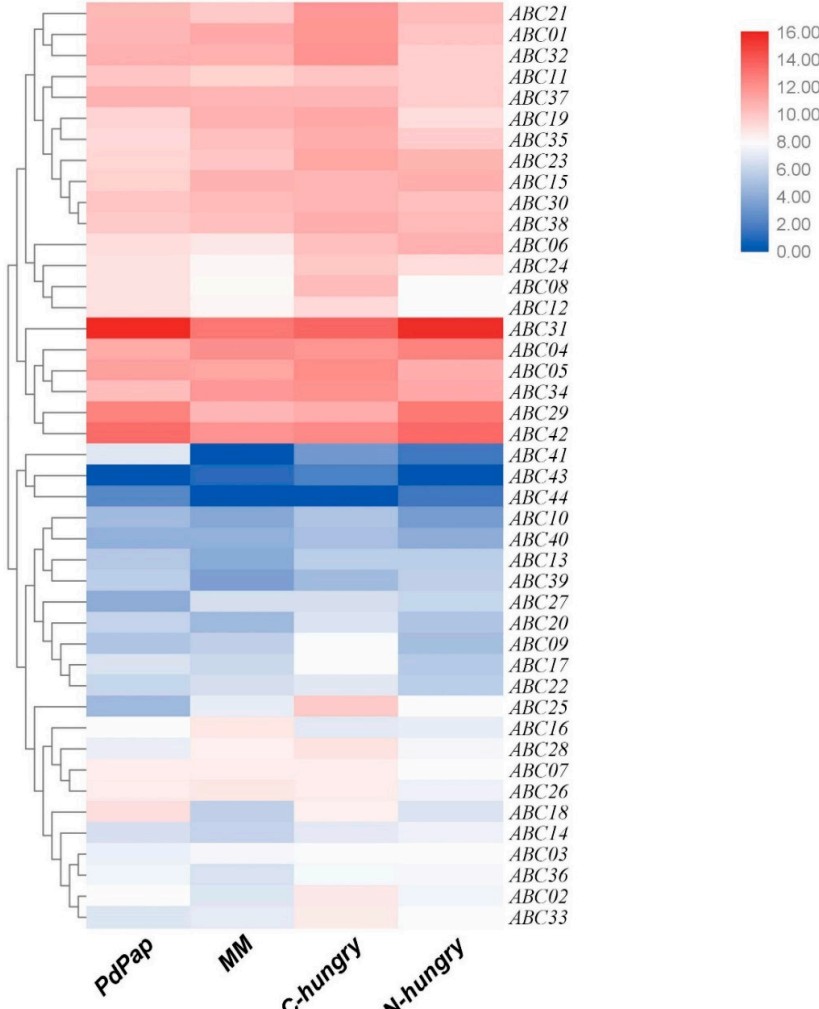

**Figure 5.** Heatmap of expression of 44 ABC transporters under four different conditions based on RNA-Seq. C-hungry = MM with 0.5% (*w/v*) ammonium sulfate, N-hungry = MM with 0.5% (*w/v*) glucose, and PdPap = variable carbon source in MM containing either 1% (*w/v*) root powder, 1% (*w/v*) stem powder, or 1% (*w/v*) leaf powder from PdPap. The heatmap of genes value = $\log_2$(RPKM), where red means high expression and blue means low expression, and the genes are clustered according to their expression.

*ABCs* highly expressed in all four conditions are mainly involved in three functions; eleven in multiple compound efflux, belonging to 3a0123 (*ABC37*), ATM1 (*ABC32* and *ABC38*), CFTR (*ABC04* and *ABC11*), MdlB (*ABC34*), MDR (*ABC21* and *ABC23*), and MRP (*ABC01*, *ABC05*, and *ABC 15*) multidomain classes; two in transport of sterols belonging to the "Protein White" multidomain class (*ABC19* and *ABC30*); and four in regulating transla-

tion belonging to the Uup multidomain class (*ABC29*, *ABC31*, *ABC35*, and *ABC 42*). These genes were thought to be involved in the stress response and should be further investigated.

### 3.7. Differential Expression of Six ABCs in Response to A. alternata CFCC82114 Toxins

To further investigate the function of ABCs under fungal phytopathogen *A. alternata* CFCC82114 toxin stress, RT-qPCR analysis was performed at six ABCs using total RNA extracted from *T. asperellum* ACCC30536 (Figure 6). The genes were chosen in combination with the comprehensive consideration of expression in transcriptome, functional prediction, and structure. In multidomain "Rim protein", "Protein White", and "Uup", we chose one gene per multidomain based on the relatively high transcription level in four transcriptomes; the proteins in other multidomains were identified and shared similar structures and predicted functions, so three encoded genes highly expressed in four transcriptomes were chosen. The six genes analyzed by RT-qPCR were *ABC01* (MRP), *ABC02* (Rim protein), *ABC04* (CFTR), *ABC05* (MRP), *ABC30* (Protein White), and *ABC31* (Uup). The results show that *ABC01*, *ABC04*, *ABC05*, and *ABC31* were upregulated after induction by the toxin, with peak $2^{3.13}$-, $2^{2.54}$-, $2^{4.92}$-, and $2^{5.09}$-fold increases at 48, 6, 6, and 6 h, respectively (Figure 6A,C,D,F). Even though *ABC02* was obviously upregulated in the treatment group at 24 and 48 h, it was also obviously upregulated in the control group (Figure 6B), so the induction of *ABC02* by toxin is not evident. A peak $2^{0.80}$-fold decrease in the transcription level at 6 h and $2^{0.29}$-fold increased transcription level at 48 h. *ABC30* was upregulated in the control group at 48 h but mostly downregulated in the treatment group (Figure 6E), with peak $2^{0.59}$-fold increased and $2^{1.83}$-fold decreased transcription levels at 6 and 48 h, respectively. The difference between the control and treatment groups is significant ($p < 0.05$), so *ABC30* was also obviously changed by the exposure to toxin. These results show that *ABC01*, *ABC04*, *ABC05*, *ABC030*, and *ABC31* are genes potentially involved in responding to pathogens.

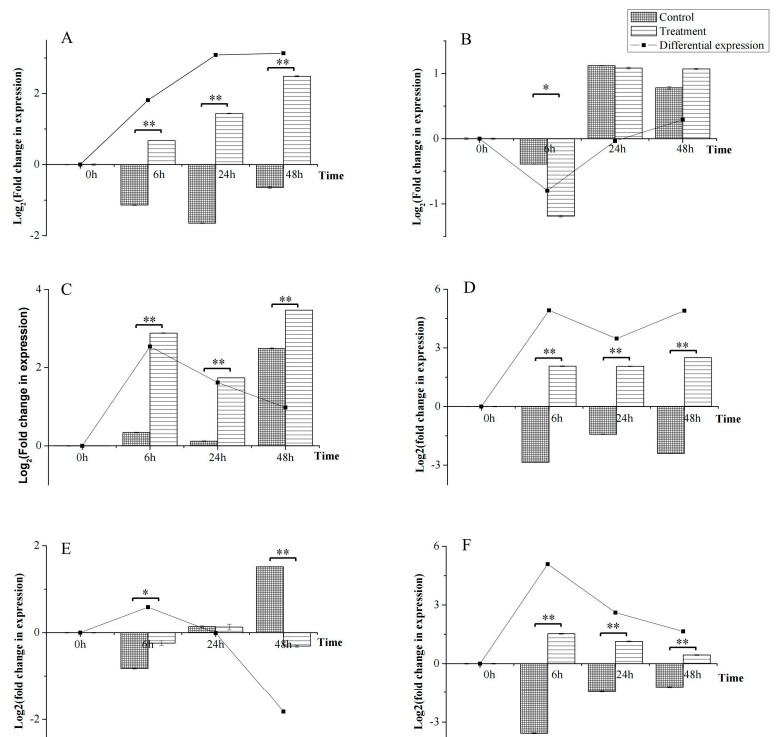

**Figure 6.** Transcription analysis of six *ABC* genes. Expression of ABC protein in *Trichoderma asperellum* cultured in MM medium (Control) or toxin-induced medium, MM with 10% fermentation filtrate of *Alternaria alternata* (Treatment). Differential expression = Treatment − Control. Error bars represent standard deviation. (**A**): *ABC01*; (**B**): *ABC02*; (**C**): *ABC04*; (**D**): *ABC05*; (**E**): *ABC30*; (**F**): *ABC31* (* $p < 0.05$, ** $p < 0.01$, n = 3).

## 4. Discussion

We found that the order of conserved domains in the ABC sequences is related to the multidomain, structure, and function (Table 2, Figure 3, Supplementary Table S1). Using this observation, we were able to rapidly predict other characteristics of the ABCs.

Multiple sequence alignment (Figure 1) revealed the sequence characteristics of 79 NBDs within the 44 ABC transporters in *T. asperellum*, which corresponds to the results for previous work on bacteria and yeast ABCs [22] or plant ABCs [22]. However, we identified mutations in some positions of all three motifs. In previous studies [9,22,23], the Walker A, ATP-binding, and Walker B motifs were "G-X(4)-G-K-[ST]", "[LIVMFY]-S-[SG]-G-X(3)-[RKA]-[LIVMYA]-X-[LIV-MF]-[AG]", and "[hydrophobic](4)-[D]", respectively. Furthermore, based on the 79 NBDs in *T. asperellum* 44 ABCs, novel Walker A, ATP-binding, and Walker B motifs were obtained, namely "G-XX-[GA]-X-G-K-[ST]", "[LIVMFYC]-S-[SGQDEVAHL]-G-X(3)-[RKALV]-[LIVMYATC]-X-[LIVMF]-[AGCT]", and "[ΦHY]-[ΦCY]-[ΦC]-[ΦCT]-[D]-[DEI]" (Φ = any hydrophobic amino acid), respectively. These findings can help us better understand NBDs and achieve better characterization of these domains in the future. Interestingly, we found that the characteristics of sequences correspond with the protein multidomain and where the NBD is located in the protein chain. This suggests that the characteristics of ABC NBDs from different protein families or that their involvement in different functions varies. Meanwhile, there was clustering when the NBD was in the same location, suggesting the NBDs in different positions have different sequence characteristics. This caused us to hypothesize that NBDs in different positions have different functions, but this theory needs further investigation.

In the phylogenetic analysis (Figure 2), evolutionary relationships between 44 ABC proteins were investigated. These 44 ABC proteins were divided into six clades, and their distribution in the phylogenetic tree is closely related to their multidomain. ABCs in the same clade belong to similar multidomain types and are likely involved in similar functions. In contrast, proteins in different clades might perform different functions or play roles in different pathways. Furthermore, proteins belonging to the "CFTR" multidomain may be more closely related to those of the "MRP" multidomain, and they may share similar mechanisms and properties, while those in "MDR" multidomain class proteins appear to be more distinct. The proteins in "Protein White" were not strongly clustered in the same clade, suggesting that even though they are similar, they may have evolved from different ancestral protein types and may take part in different functions.

The 3D structures prediction showed that the 33 transporters in ABC-3D-1 and ABC-3D-2 represent 75% of all 44 ABCs. Proteins in ABC-3D-1 and ABC-3D-2 classes belong to different multidomains but share a similar structure. The similar structure also indicates a similar mechanism in transporting compounds, which has previously been investigated: the two half structures enclose a central pocket/cavity containing multiple discrete binding sites for ligands, drugs, and transporting substrates [24,25]. Interestingly, the structure of ABC25 is clearly different from that of ABC-3D-1, with both an inward-facing and outward-facing TMD. Even though the multidomain, order of conserved domains, and predicted function are not different from other proteins in ABC-3D-1, it may have unique functions, and this requires further investigation. Two "Rim protein" members were found to have a structure (ABC-3D-3) similar to those reported to be involved in transporting sugars [26]. Four "Protein White" proteins were found to have a structure (ABC-3D-4) similar to those reported involved in transporting sterols [27]. The five ABCs in ABC-3D-5 and ABC-3D-6 all belong to Uup multidomain, each of which has more than one NBD, and these NBDs are separated by a 1–80 residue linker. Their other structural characteristics are similar to those described previously for "Uup" proteins [28]. The "Uup" proteins are reportedly involved in protein synthesis and regulating translation [29]. One mechanism to regulate translation is controlling the tRNA binding site or tRNA exit site of ribosomes [30]. This suggests that "Uup" ABC proteins may be involved in similar functions.

The promoter reflects what can activate the expression of a gene, which in turn helps us understand the function in which the gene may be involved. In this study, many of the

motifs are involved in stress responses upstream of ABC transporter genes (Figure 4). These motifs are closely related to metabolism, stress responses, and cellular defense, suggesting ABC transporter genes in *T. asperellum* may play a role in resistance and enhance survival in harsh environments. "ATM1", "Protein White", and "Uup" ABCs have relatively more motifs and may be more sensitive to environmental stimuli. In addition, numerous HSTF type motifs are located in the "Protein White" multidomain, and numerous ADR1 and GCR1 type motifs were found in genes belonging to the "MdlB" multidomain, further indicating that these genes are highly sensitive to changes in environmental conditions. In particular, GCR1, ADR1, and HSTF are closely related to G-proteins and their receptors, and since G-protein receptors act as sensors for oligopeptides, this suggests ABCs in "ATM1" and "Protein White" may be more sensitive when it comes to detecting phytopathogens based on the oligopeptides that they secrete [31]. This sensing likely stimulates the synthesis and secretion of secondary metabolites and the expression of disease-resistant genes in *Trichoderma*. Therefore, promoter analysis provides guidance on what role ABC transporter genes may play in the defense responses of *T. asperellum*.

The differential expression of 44 *ABC* transporter genes was measured through RT-qPCR in four different culture conditions: MM (control), C-hungry (simulating competing with other fungi), N-hungry (simulating interacting with other fungi), and PdPap (simulating interacting with plants) medium [31], showing that 13 of 39 predicted transport *ABCs* and 4 of 5 predicted translational regulation *ABCs* may be related to these stress responses (Figure 5). A previous study showed that proteins belonging to the multidomain "Uup" regulate translation upon nutrient starvation and depletion [32]. Our results are consistent with this observation; 4 of 5 Uup proteins were significantly upregulated under starvation conditions. Interestingly, of the highly expressed proteins, eleven are considered to be involved in multiple compound transport and belong to ABC-3D-1 and ABC-3D-2 subclasses, suggesting *ABC* genes in these two subclasses may be particularly important when dealing with pathogens and plants. We preliminarily and roughly filtered potential ABCs related to biocontrol through three simulated conditions, and these potential candidates should be further investigated in detail.

Leaf blight caused by *A. alternata* is a serious leaf disease in many plant species and needs to be well controlled. *Trichoderma* species are necrotrophic or destructive mycoparasites that can kill *A. alternata* and have been already used as useful biocontrol agents [32]. For further investigation, six genes were subjected to RT-qPCR analysis under phytopathogenic toxin stress of *A. alternata* (Figure 6). Genes *ABC01* (encoded a MRP protein, MRP), *ABC05* (MRP), and *ABC31* (Uup) were upregulated in the treatment group, while they were all downregulated in the control group, which suggests that they participate in detoxification and can help *T. asperellum* ACCC30536 survive in an environment full of other pathogens. *ABC04* (CFTR) was upregulated in the treatment group, but also in the control group, which suggests that it may be involved in functions other than detoxification. During structure prediction, we found that ABC31 only has NBDs without transmembrane, and in a previous study, it was shown that the protein only has NBDs and can increase the resistance of *Streptomyces antibioticus* to oleandomycin [33]. Thus, we hypothesize that the highly expressed gene *ABC31* may be very important in toxin resistance. In *Escherichia coli*, Uup protein can bind DNA [34] and may be involved in the excision of transposons [35]. The expression of *ABC31* (Uup) was downregulated in the control, suggesting it is suppressed in general. When *Trichoderma* encountered toxins, however, ABC31 was rapidly induced, with the highest expression value at 6 h, so we hypothesize that it is involved in responding to stress as an early key regulator. The expression of *ABC02* (Rim protein) is not obviously changed under toxin stress, but it was also highly expressed in the control group, as seen in the combined transcriptome data, which may indicate that in Rim protein genes are involved not in fungal phytopathogen toxin stress responses but are involved in some regular functions. The mechanism of ABCs in exporting substrates was due to ABCs functioning independently or with the help of a cofactor (such as membrane fused protein, outer-membrane factor, and inner-membrane factor, etc.) in exporting substrates. When the

multiple discrete binding sites in the central pocket/cavity formed by two TMDs recognize their substrate, NBDs hydrolyze ATP, produce energy, and change conformation, leading to a change in the conformation of TMDs, which ultimately leads to the substrate being transferred to the extracellular milieu [24,25,36,37]. In this study, how the upregulated genes participate in detoxification, what exact components in the fermentation liquid of *A. alternata* are exported by the ABCs, and how these components bind to the ABCs should be further investigated. The findings indicate reference genes for genetic engineering, with which we can construct transgenic strains for further analysis and expect to obtain a strain with good biocontrol ability that cannot easily suppressed by toxins.

In conclusion, conserved domain prediction showed that ABC proteins in *T. asperellum* can be divided into nine groups based on their multidomains: MRP, CFTR, MdlB, ATM1, MDR, Protein White, Rim protein, 3a0123, and Uup. The NBD sequence characteristics, phylogenetic relationship, and structures are correlated with the multidomain. Numerous motifs related to stress responses were also found in their promoter regions. In addition, transcriptome analysis showed that when *T. asperellum* ACCC30536 was under pathogenic and botanic stress, 17 of 39 predicted transport ABCs and 4 of 5 predicted translational regulation ABCs had higher expression. In further RT-qPCR analysis, *ABC01*, *ABC04*, *ABC05*, *ABC30*, and *ABC31* were found to be significantly altered. They are involved in the response to pathogenic *A. alternata* toxin stresses and, hence, are worthy of further study. In this study, the properties of ABC proteins in *T. asperellum* were characterized, which lays the foundation for understanding how they help *T. asperellum* survive under stress, and we identified potential ABC proteins that may be involved in toxin stress responses, which is also important for revealing the biocontrol mechanism of *Trichoderma*.

## 5. Disclosure

All experiments undertaken in this study comply with the current laws of the country in which they were performed.

**Supplementary Materials:** The following supporting information can be downloaded at: https://www.mdpi.com/article/10.3390/cimb45020101/s1, Table S1: Properties of the 44 ABC transporters in *Trichoderma asperellum*.

**Author Contributions:** Z.-Y.Y., H.-Y.D., Y.-Z.Z. and K.L. performed the experiments, analyzed the data, and were major contributors in writing the manuscript. H.-Y.D., Y.-Z.Z., K.L., S.C. and X.G. collected the data. P.-W.G., Z.-Y.W. and Z.-H.L. provided instruments and checked the experiments and manuscript. All authors have read and agreed to the published version of the manuscript.

**Funding:** This work is supported by grants from the National Natural Science Foundation of China (NSFC:31870627), Startup Funds of Talent Introduction of Ningxia University (030900002218), and the National High Technology Research and Development Program (the 13th Five-Year Plan Program) (grant number 2016YFC0501505).

**Institutional Review Board Statement:** Not applicable.

**Informed Consent Statement:** Not applicable.

**Data Availability Statement:** The datasets generated and analyzed during the current study are available in the JGI Genome Portal repository (http://genome.jgi.doe.gov/pages/tree-of-life.jsf., accessed on 3 March 2021 and https://phytozome.jgi.doe.gov/pz/portal., accessed on 3 March 2021) and the National Center for Biotechnology Information (NCBI) repository (https://blast.ncbi.nlm.nih.gov/Blast.cgi, accessed on 3 March 2021).

**Conflicts of Interest:** The authors declare no conflict of interest.

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
