# Peer review of "Analysis of the Properties of 44 ABC Transporter Genes from Biocontrol Agent Trichoderma asperellum ACCC30536 and Their Responses to Pathogenic Alternaria alternata Toxin Stress"

_cimb, doi:10.3390/cimb45020101_

Round 1

Reviewer 1 Report

Well written manuscript

Author Response

Dear Reviewer:

Thank you for your advice, please see the attachment to review our modification.

Sincerely,

Ze-Yang Yu

Reviewer 2 Report

The manuscript is very articolate and full of information, this type of study needs to be well explained. I found it difficult to read and very confusing. I cannot understand how your study can assist the development of environmentally friendly biocontrol agents, perhaps this point needs to be better explained in the Discussion, there are no links between the results obtained and the main aim of the study. The English needs to be reviewed. There are many inaccuracies, for example in line 142, the sentence is understandable; furthermore, you chose six genes for RT-qPCR but in Table 1 you list seven of them. The manuscript can be reconsidered after major revision.

Author Response

Review 2:-I found it difficult to read and very confusing. I cannot understand how your study can assist the development of environmentally friendly biocontrol agents, perhaps this point needs to be better explained in the Discussion, there are no links between the results obtained and the main aim of the study.

Response: According to the advice, we have modified the sentence in Introduction section to make it clear, and we have added description in Discussion section as below.

In Introduction Section, We have modified the sentence to “The findings expand our understanding of Trichoderma tolerance to pathogenic toxins or phytotoxins, and the characteristics of these potential genes can serve as a reference for identifying ABC genes or finding robust ABC genes in other Trichoderma species or organisms, which could ultimately assist in the development of improved environmentally friendly biocontrol agents.” in line 102-106 of revised manuscript.

In Discussion Section, we have added the description “The findings indicate reference genes for genetic engineering, with which we can construct transgenic strains for further analysis and expect to obtain a strain with good biocontrol ability that cannot easily suppressed by toxins.” in line 468-471 of revised manuscript. 

  • - The English needs to be reviewed.

Response: According to the advice, the revised manuscript has been edited by a native English speaker.

  • - There are many inaccuracies, for example in line 142, the sentence is understandable; furthermore, you chose six genes for RT-qPCR but in Table 1 you list seven of them..

Response: The language has been checked by a native English speaker, which may improve the language and make the sentence clear. We have deleted the excess primers (ABC 42 ) in Table 1.

Reviewer 3 Report

I have read and evaluated the Ms entitled “The property analysis of 44 ABC transporter genes from bio- to pathogenic Alternaria alternata toxin stress Hua-Ying Du1#, Yu-Zhou Zhang2#†, Kuo Liu1, Pei-Wen Gu1, Shuang Cao1, Xiang Gao1, Zhi-Ying Wang2, Zhi-Hua Liu2,3, Ze-Yang Yu1,2,  I have found some lacunae which have to revise by authors. Some works have been given  and theses should be performed honestly before acceptance of this Ms for publication

1.       In line 90-91 Authors have written that :How- ever, there are currently no studies on ABC transporters in other Trichoderma species, -------”    But after internet search there are works on ABC transporters of other sp of Trichoderma  . For sample paper like :  He LiuMing ChengShanshan ZhaoCongyu LinJinzhu Song, and Qian Yang* ATP-Binding Cassette Transporter Regulates N,N′-diacetylchitobiose Transportation and Chitinase Production in Trichoderma asperellum T4. Int J Mol Sci. 2019 May; 20(10): 2412.Published online 2019 May 15. doi: 10.3390/ijms20102412.

2.       It is quit okay that ABC transporters  in T. asperellum have upregulated  due to toxic or harsh environment  and T. asperellum has  ABC Transporter to fight the toxic  substance  secreted by A. alternata. Kindly explain the ligand – toxin ( of  A. alternata )  binding mechanism in ABC transporters of T. asperellum in discussion section.

3.       Kindly mention  kind of mycoparasitism of  T. asperellum   against A. alternata during discussion. .  Is it  Necrotrophic or bio tropic mycoparasitism ???.

4.        Authors mentioned that  culture filtrate of  A. alternata is toxic substance  but have they detected and identified the specific toxic ??? Toxins may be multiple , but which toxin induced  ABC transporter genes is my question. Authors  have broadly  supposed the toxin is present in culture filtrate of A. alternata . Authors  should at least do one toxin assay.  Furthermore , as a positive control,  Author must have to take a pure A. alternata toxin in the experiment of ABC transporter gene expression by RT- PCR.  For sample paper: Jianan Sun, Tailong Zhang, Yaqian Li, Xinhua Wang, Jie Chen ( 2019) Functional characterization of the ABC transporter TaPdr2 in the tolerance of biocontrol the fungus Trichoderma atroviride T23 to dichlorvos stress , Biological control 129: 102-108 .

5.        Another think is that don’t mixed the idea of hunger or nutrient deficit condition or environment  of T. asperellum  with toxic environment  of T. asperellum. Two conditions induce    ABC transporter gene expression differently, and we have to find precise ABC transporters gene expression separately. “It is suggested  that fungal adaptation often requires a selected number of core genes to regulate a particular mycoparasitic lifestyle. A comparative analysis of ABC family genes reveals critical differences between various microparasitic lifestyles, while also pinpointing distinctive evolutionary advantages of S. mycoparasitica-specific mycoparasitism, thus, opening new directions for future research on biocontrol of mycotoxigenic Fusarium pathogens. ”Kim etal 2022.  So, authors have to explain in discussion.  I can mention here : “In mycoparasite Clonostachys rosea, abcG5, belonging to the PDR family, seems to be an essential ABC transporter gene responsible for antagonistic and biocontrol activity [27]. Interestingly, abcG5 gene expression was induced by mycotoxin zearalenone [28], while its deletion in C. rosea resulted in lowering fungal tolerance to zearalenone produced by plant pathogenic Fusarium species [27].” Kim et al  2022    27. Dubey, M.K.; Jensen, D.F.; Karlsson, M. An ATP-binding cassette pleiotropic drug transporter protein is required for xenobiotic tolerance and antagonism in the fungal biocontrol agent Clonostachys rosea. Mol. Plant Microbe Interact. 2014, 27, 725–732.         28. Kosawang, C.; Karlsson, M.; Jensen, D.F.; Dilokpimol, A.; Collinge, D.B. Transcriptomicprofiling to identify genes involved in Fusarium mycotoxin deoxynivalenol and zearalenone tolerance in the mycoparasitic fungus Clonostachys rosea. BMC Genom. 2014, 15, 55.             Kim, S.H.; Vujanovic, V. ATP-Binding Cassette (ABC) Transporters in Fusarium Specific Mycoparasite Sphaerodes mycoparasitica during Biotrophic Mycoparasitism. Appl. Sci. 2022, 12, 7641. https://doi.org/10.3390/ app12157641

6.        I found numerous grammatical and spelling mistakes in whole  Ms: I am giving some sample mistakes i.  Correct the sentence “may important when Trichoderma survive---" ii. Line 23  correct the spelling “ envirionment “  iii. Please read the sentence  where past tense and present tense are mixed. Line 24-27  “In this study, genome searches revealed 44 ATP-binding cassette (ABC) transporters encoded in the genome of Trichoderma asperellum and the 44 ABC transporters were divided into six types based on three dimensional (3D) structure prediction, of which four types, containing 39 ABCs, are considered involved in transport, and two types, containing 5, are considered involved in regulating translation.” Kindly correct the sentences.  Iv. Line 32 :” interacte”, kindly correct the spelling mistake.  V.Line 46 : authors used the term “fungicide”   but it should be fitted best as  “biofungicide”  vi. Line  142 correct the spelling “whther” vi. Line 389 : Correct this sentence  “Promoter reflect what can active the expression of gene which help us understand what function the gene involved” vii.  Correct the sentence :Line 410  -------translation regulation ABCs may be relate to these stress response. So , authors must do line to line grammatical and spelling  checking and correction.

Author Response

(The authors gave the same response as above.)

Round 2

Reviewer 2 Report

The authors have greatly improved the manuscript, now it can be accepted in present form

Reviewer 3 Report

I have read attentively the revised Ms.  Authors have gave significant value of reviewers suggestions. They have  modified or revised the Ms as per their capacity. English language has been  checked and  corrected. Now the Ms may be published in this journal.